# mRNA induced expression of human angiotensin-converting enzyme 2 in mice for the study of the adaptive immune response to severe acute respiratory syndrome coronavirus 2

**Mariah Hassert**[1], **Elizabeth Geerling**[1], **E. Taylor Stone**[1], **Tara L. Steffen**[1], **Madi S. Feldman**[2], **Alexandria L. Dickson**[1], **Jacob Class**[3], **Justin M. Richner**[3], **James D. Brien**[1]*, **Amelia K. Pinto**[1]*

1 Department of Molecular Microbiology and Immunology, Saint Louis University, St. Louis, Missouri, United States of America, 2 Department of Biomedical Engineering, Saint Louis University, St. Louis, Missouri, United States of America, 3 Department of Microbiology and Immunology, University of Illinois College of Medicine, Chicago, Illinois, United States of America

* james.brien@health.slu.edu (JDB); amelia.pinto@health.slu.edu (AKP)

## Abstract

The novel human coronavirus, severe acute respiratory syndrome coronavirus 2 (SARS-CoV-2) has caused a pandemic. Critical to the rapid evaluation of vaccines and antivirals against SARS-CoV-2 is the development of tractable animal models to understand the adaptive immune response to the virus. To this end, the use of common laboratory strains of mice is hindered by significant divergence of the angiotensin-converting enzyme 2 (ACE2), which is the receptor required for entry of SARS-CoV-2. In the current study, we designed and utilized an mRNA-based transfection system to induce expression of the hACE2 receptor in order to confer entry of SARS-CoV-2 in otherwise non-permissive cells. By employing this expression system in an *in vivo* setting, we were able to interrogate the adaptive immune response to SARS-CoV-2 in type 1 interferon receptor deficient mice. In doing so, we showed that the T cell response to SARS-CoV-2 is enhanced when hACE2 is expressed during infection. Moreover, we demonstrated that these responses are preserved in memory and are boosted upon secondary infection. Importantly, using this system, we functionally identified the CD4+ and CD8+ structural peptide epitopes targeted during SARS-CoV-2 infection in H2$^b$ restricted mice and confirmed their existence in an established model of SARS-CoV-2 pathogenesis. We demonstrated that, identical to what has been seen in humans, the antigen-specific CD8+ T cells in mice primarily target peptides of the spike and membrane proteins, while the antigen-specific CD4+ T cells target peptides of the nucleocapsid, membrane, and spike proteins. As the focus of the immune response in mice is highly similar to that of the humans, the identification of functional murine SARS-CoV-2-specific T cell epitopes provided in this study will be critical for evaluation of vaccine efficacy in murine models of SARS-CoV-2 infection.

**Data Availability Statement:** All relevant data are within the manuscript and its Supporting Information files.

**Funding:** This work was supported by Saint Louis University COVID-19 research Seed Funding awarded to JDB and AKP and National institutes of Health grant F31 AI152460-01 from the National Institute of Allergy and Infectious Diseases (NIAID) awarded to MH. The funders had no role in study design, data collection and analysis, decision to publish, or preparation of the manuscript.

**Competing interests:** The authors have declared that no competing interests exist.

## Author summary

The development of tractable small animal models is critical to gain an understanding of the immune response to the novel human coronavirus, severe acute respiratory syndrome coronavirus 2 (SARS-CoV-2), and for the evaluation of vaccines against the virus. However, the development of murine models of infection has been hindered due to the lack of expression of the human angiotensin-converting enzyme 2 (hACE2), which is the receptor required for entry of SARS-CoV-2. In this study, we cloned the hACE2 gene into an mRNA expression vector and demonstrated that transfection with this mRNA allowed for SARS-CoV-2 entry and replication. We utilized this novel method of hACE2 expression in mice by in vivo mRNA transfection to characterize the adaptive immune response to SARS-CoV-2. This unique and tractable model allowed for the first ever characterization of the murine SARS-CoV-2 specific T cell response. This information will be critical to determining the correlates of protection against the virus and for the evaluation of vaccines.

## Introduction

The pandemic level spread of severe acute respiratory syndrome coronavirus 2 (SARS-CoV-2), and the resulting outbreak of coronavirus disease 2019 (COVID-19) drives a need for the development of a range of novel animal models to understand the virus induced pathology and evaluate vaccines and therapeutics. SARS-CoV-2 results in a range of human disease phenotypes, most likely through different mechanisms [1]. So, while each pre-clinical model will have its strengths and weaknesses, the heterogeneity of the human disease propels the need for a range of pre-clinical animal models and tools. These animals and tools will be necessary for the evaluation of direct acting therapeutics, host targeted therapeutics, the identification of vaccine correlates of protection, and a fundamental understanding of the drivers of pathology and disease.

The genome of the SARS-CoV-2 reference strain (Wuhan-Hu-1) (NC_045512.2) is ~ 30 Kb and encodes four main structural genes: Spike, envelope (Env), membrane (Mem), and nucleocapsid (NC), as well as 16 nonstructural proteins (nsp1-16) and multiple accessory proteins [2]. In humans, T cell epitope analysis suggests that the majority of the CD4$^+$ and CD8$^+$ T cell responses identified in SARS-CoV-2 positive patients are directed toward the spike with responses also detected against Mem and NC [3,4]. Multiple studies have also shown following the resolution of SARS-CoV-2 infection there is a strong neutralizing IgG antibody response detected against the receptor binding domain (RBD) of spike [5–11]. These studies support the development of vaccines that induce strong responses to the SARS-CoV-2 spike protein as they will induce T cell as well as antibody responses to the virus. What is currently needed are preclinical animal models, which can evaluate the efficacy and immunogenicity of the current vaccines as well as determine if the immune response generated against SARS-CoV-2 infection supports a protective or pathogenic role in COVID-19.

From the study of multiple human and animal CoVs, we know that the interaction between the spike glycoprotein and its cognate receptor is one of the main determinates regarding species and cellular tropism [12]. For HCoV-NL63, SARS-Co-V and SARS-CoV-2, the cognate receptor is the angiotensin-converting enzyme 2 (ACE2) [12–17]. ACE2 has been shown to be expressed in lungs, heart, kidneys and intestine and primarily functions as an enzyme controlling the maturation of angiotensin [18,19]. Mice also express ACE2, and an amino acid

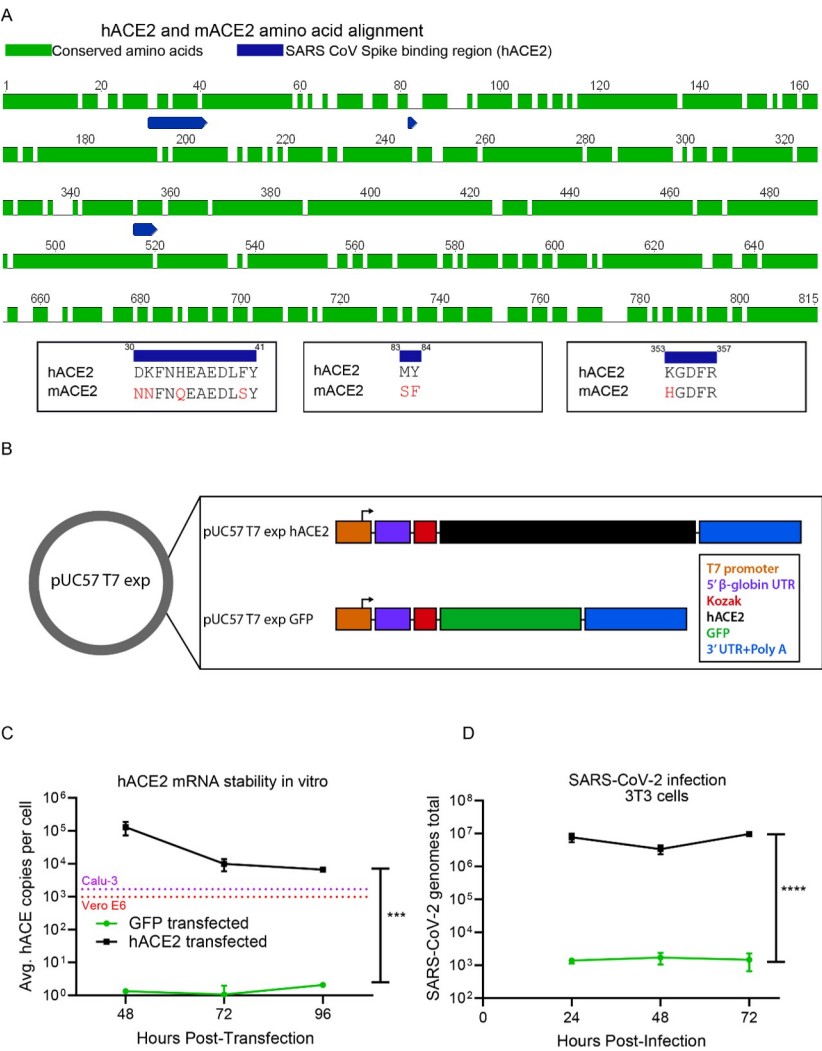

**Fig 1. Transient expression of hACE2 in murine cells allows for SARS-CoV-2 entry.** (**A**) Amino acid identity between hACE2 and mACE2. The amino acid sequences of human ACE2 (hACE2) (BAB40370) and murine ACE2 (mACE2) (NP_081562) were globally aligned using a BLOSUM62 cost matrix in the computational program Geneious. The two sequences showed 81.2% amino acid identity. Specific regions of interest included the amino acid residues important for SARS-CoV spike binding (and putative SARS-CoV-2 spike binding) (residues 30–41, 83–83, and 353–357). Multiple amino acid differences were noted in red in these critical regions between hACE2 and mACE2. (**B**) mRNA expression construct for induced expression of hACE2. T7 expression cassettes for hACE2 (and negative control GFP) were cloned into a pUC57 backbone by Gibson assembly. Each expression cassette includes a T7 promoter element, a 5' β-globin UTR, Kozak sequence, CDS of each gene of interest (hACE2 or GFP), followed by a 3' UTR and polyA tail. Following plasmid linearization and purification, mRNA was prepared in vitro using an ARCA T7 in vitro transcription kit. (**C**) In vitro stability of hACE2 mRNA. 2x10$^6$ Murine 3T3 cells were transfected with 2 μg of either hACE2 or GFP mRNA and plated at a density of 5x10$^6$ cells in each well of a 6 well dish. At 48, 72, and 96 hours post transfection, the stability of the hACE2 mRNA within the cells was assessed by qRT-PCR. Statistical significance was determined by 2-way ANOVA (p = 0.004). (**D**) In vitro expression of hACE2 permits SARS-CoV-2 entry. 2x10$^6$ Murine 3T3 cells were transfected with 2 μg of either hACE2 or GFP mRNA and plated at a density of 5x10$^6$ cells in each well of a 6 well dish. 24 hours post transfection, cells were infected with SARS-CoV-2. At 24, 48, and 72 hours post infection, SARS-CoV-2 RNA was quantified from the 3T3 cell pellets by qRT-PCR. Statistical significance was determined by 2-way ANOVA (p<0.0001).

alignment of the murine ACE2 (mACE2) with human ACE2 (hACE2) shows an approximate 81 percent identity between the two proteins (**Fig 1A**). As was shown for SARS-CoV [20], specific interactions between the spike glycoprotein, specifically the receptor binding motif of

SARS-CoV-2 and hACE2, likely explain why SARS-CoV-2 infection of wild type mice does not occur [8]. Therefore, to establish a susceptible mouse model to study pathogenesis and immune responses to SARS-CoV-2, we must either alter the SARS-CoV-2 virus to recognize the mACE2 [21] or express the hACE2 in mice [2,8,22–27]. Various strategies have been employed by multiple groups to facilitate the expression of hACE2 in mice, from CRISPR/Cas9 mediated hACE2 transgenics [2,22,24–27], to adenovirus [8] or adeno associated virus (AAV) expressing hACE2 [23]. SARS-CoV-2 intranasal infections in these systems have achieved detectable virus in the lungs and trachea leading to viral pneumonia histologically [8,23,28]. Alternatively, targeted mutagenesis of the SARS-CoV-2 spike protein, facilitating enhanced interactions with murine ACE2, results in a similar phenotype [29]. Golden Syrian hamsters also offer a potential avenue to explore vaccines and therapeutics against SARS-CoV-2, and multiple groups have found that SARS-CoV-2 efficiently replicates in hamsters [21,30]. However due to the lack of reagents at this time, hamster models have had limited utility in the studies of immune correlates of protection from infectious diseases [31].

By developing an mRNA delivery platform to express hACE2 we were able to permit SARS-CoV-2 entry and replication into receptor negative cells. The transfected RNA is highly durable, as we were able to detect mRNA expression in culture at high levels for at least four days following transfection. Translating this model in vivo, in C57BL/6 type 1 interferon receptor deficient (Ifnar1$^{-/-}$) mice, we were able to demonstrate hACE2 transfection in approximately 20% of cells in the lungs and liver. Through infection of mice transfected to express the hACE2 receptor, we were able to rapidly identify and characterize both neutralizing antibodies and virus specific T cell responses generated against SARS-CoV-2. We noted many significant benefits of the mRNA transfection system; primarily the tractability and speed of a novel system to study adaptive immune responses to emerging viral infections in organisms which lack the necessary host factors for viral entry. In this study we identified nine CD8+ and six CD4+ H2$^{b}$ restricted SARS-CoV-2-specific T cell epitopes. Allmost all of the mice that express hACE2 transgenically and transiently are on an H2$^{b}$ background, including the K18-hACE2 transgenic mouse model [32]. Importantly, we confirmed our identification of the SARS-CoV-2 T cell epitopes in the K18-hACE2 transgenic model if SARS-CoV-2 pathogenesis, indicating that this information will immediately become useful for understanding the role of the antigen specific T cell responses in both protection and pathology associated with SARS-CoV-2 in most of the existing animal models of SARS-CoV-2 mediated disease.

## Results

### hACE2 mRNA construct

To generate a construct that could produce robust expression of hACE2 mRNA, we used a pUC57 vector backbone consisting of four core components: 1) a type II T7 promoter for a high transcription rate *in vitro* and the production of RNA transcripts with homogeneous 5' and 3' termini; 2) a beta globin 5'UTR for optimized mRNA expression within mammalian cells; 3) 3' UTR from alpha globin for mRNA stability and 4) A 153 base adenosine nucleotide stretch as the polyA tail. Into this backbone, we cloned either the coding sequence for hACE2 or green fluorescent protein (GFP), as a control (**Fig 1B**). mRNA from either construct was then generated using a T7 ARCA in vitro transcription reaction. To demonstrate the stability of transcribed hACE2 mRNA, murine fibroblast 3T3 cells were transfected with the in vitro transcribed and purified GFP or hACE2 mRNA. At 48-, 72- and 96-hours post-transfection, expression of hACE2 was measured by qRT-PCR (**Fig 1C**). For hACE2, the qRT-PCR data revealed that we were able to detect stable transfected hACE2 mRNA in the 3T3 cells for four days, with only a minimal decline in expression levels over that time period. For at least 96

hours post-transfection, the expression level of hACE2 in this hACE2 transfected cell line was similar to the mRNA expression of hACE2 we detected in highly SARS-CoV-2 susceptible Calu-3 and Vero-E6 cell lines (**Fig 1C**). With the cells transfected with the GFP expressing mRNA construct, we performed flow cytometry and demonstrated that we could achieve greater than 90 percent transfection efficiency in vitro, and that the GFP expression was maintained for at least 72 hours (**S1 Fig**). These results demonstrated that we were able to generate an mRNA vector expression system and deliver hACE2 to non-hACE2 expressing murine 3T3 cells, resulting in RNA levels similar to what we observed in the susceptible cell lines, Calu-3 and Vero-E6 [33] and that was stable for at least 96 hours.

We next confirmed that expression of hACE2 induced by our mRNA construct conferred susceptibility of murine cells to SARS-CoV-2 by infecting the hACE2 or GFP transfected 3T3 cells with SARS-CoV-2 (**Fig 1D**). The hACE2 or GFP mRNA transfected cells were each plated into separate 6 well plates, and 24 hours after transfection, they were infected with SARS-CoV-2 at a multiplicity of infection (MOI) of 0.01. The infected cells were harvested at 24, 48, and 72 hours post infection and the viral genomes were quantified by qRT-PCR. The 3T3 cells transfected with the hACE2 had more virus detected at every time point post infection with approximately four logs higher viral genome copies within the cells as compared to the cells that had been transfected with the GFP expressing mRNA (**Fig 1D**). These results demonstrate that the transfection of murine cells with hACE2 mRNA confers susceptibility to SARS-CoV-2 entry.

## In vivo transfection efficiency

To determine if we could induce gene expression via mRNA transfection in vivo, we first administered 10 μg of RNA prepared in Polyplus in vivo-jet RNA in vivo transfection reagent. RNA encoding either GFP or firefly luciferase (fLuc) was administered via intravenous (IV) and intranasal (IN) combination route to type I interferon receptor 1 deficient (Ifnar1$^{-/-}$) mice. 24 hours post transfection, mice were injected intraperitoneally (IP) with luciferase substrate D-Luciferin and imaged via IVIS (in vivo imaging system for bioluminescence) (**Fig 2A**). Compared to the GFP control mouse, the mouse transfected with fLuc mRNA demonstrated detectable bioluminescence likely in the liver and perhaps in the lung. This indicated to us, that our method of in vivo mRNA delivery was sufficient to detect expression of a reporter protein.

To confirm these findings, mice were transfected in the same manner with either GFP (n = 3), hACE2 (n = 8), or just vehicle (n = 3). At 24 hours post transfection, lungs and livers were harvested and either used to assess hACE2 mRNA stability via qRT-PCR (**Fig 2B**), or hACE2 transfection efficiency by flow cytometry (**Fig 2C**). At 24 hours post transfection, we detected significant amounts hACE2 mRNA in the lungs and liver of hACE2 transfected animals, indicating the relative stability of the transfected mRNA (**Fig 2B**). To determine transfection efficiency by protein expression, cellular isolates from the lungs and livers of transfected mice were stained with a human ACE2 specific antibody and analyzed by flow cytometry (**Fig 2C**). Between 14–20% of cells analyzed in the liver and lungs of hACE2 transfected mice were expressing hACE2 protein on the cell surface. Taken together, these data demonstrate that by this method of in vivo mRNA transfection, we could achieve hACE2 receptor expression in a subset of cells in vivo, which could permit SARS-CoV-2 entry.

## Adaptive immune response to SARS-CoV-2

To begin to identify the immune correlates of protection we used the hACE2 mRNA transfection system to study the adaptive immune response to SARS-CoV-2 in type I interferon receptor 1 deficient (Ifnar1$^{-/-}$) mice (**Fig 3**). We chose to use the Ifnar1$^{-/-}$ mice, because, like other

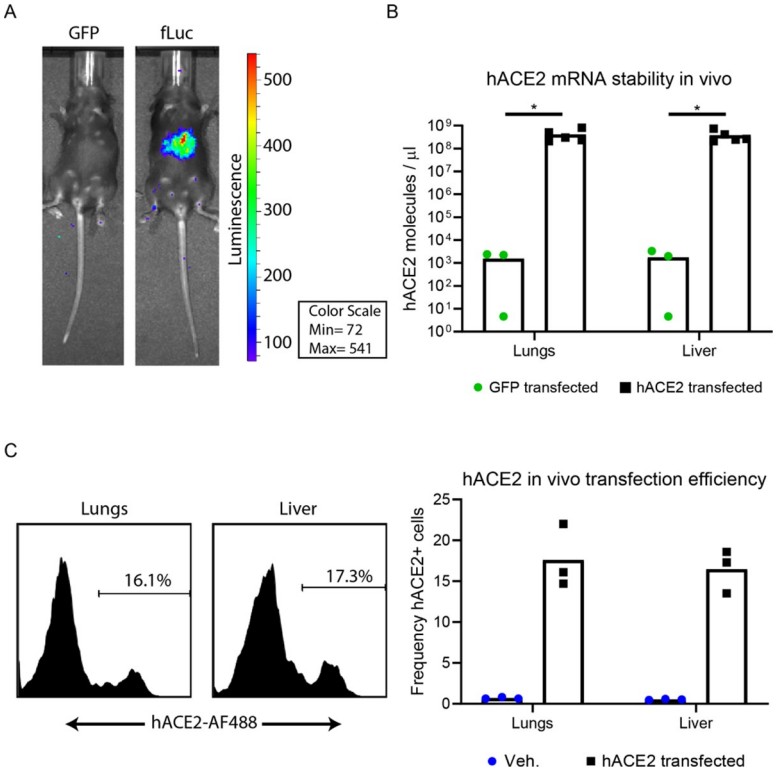

**Fig 2. In vivo transfection efficiency.** (**A**) fLuc reporter expression. 10 μg of fLuc or GFP mRNA was prepared in Polyplus in vivo-jet RNA in vivo transfection reagent and administered to Ifnar[1-/-] via IV and IN combination route. 24 hours following transfection, mice were injected IP with D-luciferin and imaged via IVIS 15 minutes later. (**B**) hACE2 mRNA stability in vivo. Ifnar1-/- mice were transfected with 10 μg of GFP or hACE2 mRNA. 24 hours post transfection, mice were euthanized and liver and lung tissue homogenized in TriReagent RT for RNA extraction. hACE2 mRNA levels were quantified from extracted RNA via qRT-PCR. Statistical significance was determined by Mann-Whitney test (p = 0.03 and p = 0.03, in the lungs and liver, respectively) (**C**) hACE2 in vivo transfection efficiency, Ifnar1-/- mice were transfected with 10 μg of hACE2 mRNA or vehicle alone. 24 hours post transfection, mice were euthanized and liver and lung tissue were dissociated into single cell suspensions and stained with a human-ACE-2 specific antibody, followed by an AF488-conjugated anti-human secondary antibody. Live cells were analyzed on an Attune focusing flow cytometer and are represented as frequency of hACE2 expressing cells.

viral infections, coronaviruses encode genes which dampen or block the type I IFN response creating a more susceptible environment for the establishment of infection [34,35]. Studies with multiple viruses have shown that the loss of IFNAR, and therefore the significant dampening of IFN stimulated gene production, results in IFNAR deficient cells being highly susceptible to viral infections [36–45].

In these studies, we followed a prime boost strategy (**Fig 3A**), similar to what would be done for a vaccine, so to enhance the detection of the immune response to SARS-CoV-2 in the Ifnar1[-/-] mice. First, the mice were transfected with 10 μg of mRNA per animal, encoding either hACE2 or GFP as a control. Twenty-four hours after mRNA in vivo delivery, we infected the hACE2 or GFP transfected Ifnar1[-/-] mice with a total of 5x10[4] focus forming units (FFU) per mouse of SARS-CoV-2 administered both intravenously (IV) and intranasally (IN). Blood was collected eight- and ten-days post infection for acute phase T cell analysis and serology, respectively. After day 29 post primary infection, the Ifnar1[-/-] mice received a second hACE2 or GFP mRNA transfection, followed 24 hours later by a boost with 5x10[4] FFU per mouse of SARS-CoV-2, administered both IV and IN route. Five days post boost, blood was collected for T cell and antibody analysis.

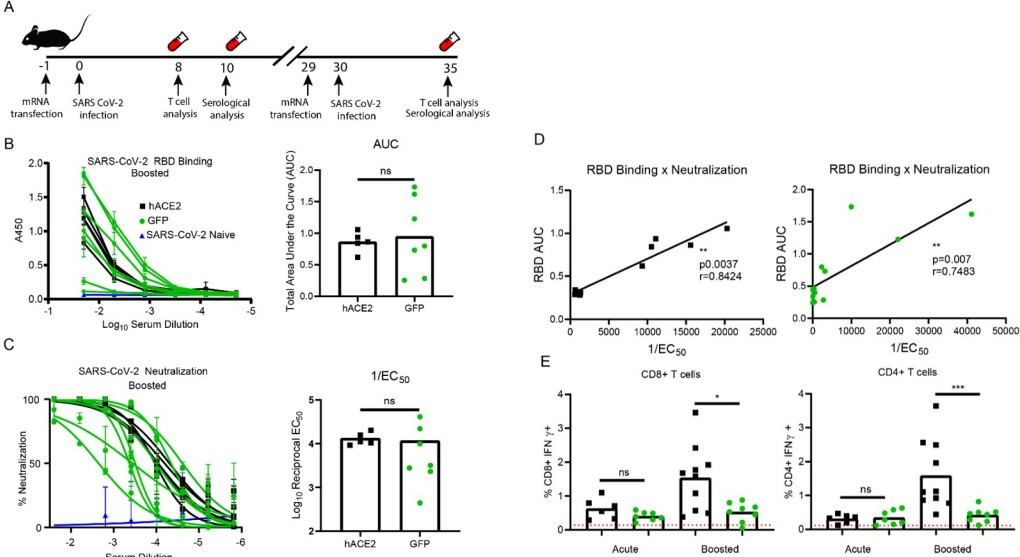

**Fig 3. In vivo transfection of hACE2 mRNA yields enhanced CD4+ and CD8+ T cell responses. (A)** Experimental design. 8-week-old Ifnar1-/- mice were transfected with 10 μg of GFP or hACE2 mRNA. 24 hours post transfection, mice were infected with SARS-CoV-2. At days 8 and 10 post infection, blood was collected for acute phase T cell stimulation assays and serology recpectively. At days 29 post initial infection, the mice were again transfected with 10 μg of GFP or hACE2 mRNA and infected 24 hours later. 5 days post boost, blood and serum was collected for memory recall T cell stimulation assays and serological analysis. **(B)** Spike RBD ELISA. Serially diluted serum from mice five days post boost was added to a recombinant SARS-CoV-2 spike RBD protein coated plate to determine RBD binding potential by absorbance at 450nm. The results of the dilution series was used to calculate the area under the curve calculation. **(C)** Neutralization potential of polyclonal sera. As above serum from boosted mice was serially diluted with ~100 focus forming units of SARS-CoV-2. Neutralization was determined by enumerating a reduction in infectious particles with increased serum concentration and determining the $EC_{50}$. **(D)** Correlation between RBD binding and SARS-CoV-2 neutralization for both the hACE2 and GFP transfected SARS-CoV-2 infected Ifnar1$^{-/-}$ mice. Correlative analysis between RBD ELISA AUC values and $1/EC_{50}$ values was performed using linear regression analysis and a two-tailed Pearson analysis $p<0.0037$, $r = 0.8424$ and $p<0.007$, $r = 0.7483$ for the hACE2 and GFP transfected mice respectively. **(E)** Global T cell responses during SARS-CoV-2 infection. Eight days post infection and five days post boost, collected blood lymphocytes were stimulated with anti-CD3 nd stained with anti CD19, CD4, CD8, IFNγ and TNFα. The frequency of responding CD8+ and CD4+ T cells was demonstrated by quantifying the frequency of CD8+ or CD4+ T cells producing IFNγ. The gating strategy is shown in **S3 Fig**. Statistical significance was determined by Mann-Whitney test ($p = 0.025$ and $0.0009$ for CD8+ and CD4+ boosted T cell responses, respectively).

To examine the antibody response directed against SARS-CoV-2, we performed an indirect ELISA against the RBD of the SARS-CoV-2 spike protein, a target of the neutralizing antibody response in both mice and humans [5–9]. To quantify the anti-RBD antibody response, we compared the polyclonal sera from naïve Ifnar1$^{-/-}$ mice, to that of the SARS-CoV-2 infected Ifnar1$^{-/-}$ mice transfected with either the hACE2 (n = 5) or GFP mRNA (n = 7) five days post boost (**Fig 3B**). We noted that only the Ifnar1$^{-/-}$ mice that had been infected with SARS-CoV-2 were able to bind RBD. Five days post SARS-CoV-2 boost all mice that had received hACE2 mRNA, and six out of seven of the GFP group had detectable RBD binding. Analysis of the area under the curve (AUC) showed there were no detectable differences between the SARS CoV-2 infected Ifnar1$^{-/-}$ mice. We noted a similar result with the ELISA analysis of the blood ten days post primary infection, where all mice showed low levels of an anti-RBD IgG response that were not significantly different between the two groups (**S2A Fig**).

While there were SARS-CoV-2 RBD specific antibodies present in both the infected GFP and hACE2 transfected mice, we were interested to determine if there was a difference in the neutralization capacity of the antibody response generated in the hACE2 transfected mice as compared to the GFP transfected controls. To test this, we performed focus reduction

neutralization (FRNT) assays on the serum samples (**Fig 3C**). The average $FRNT_{50}$ values were 1:1050 and 1:3125, for the hACE2 or GFP transfected mice respectively. Analysis of the individual neutralization curves and $FRNT_{50}$ values from the serum collected post boost showed that three of the seven GFP transfect mice neutralized SARS-CoV-2 as well as the hACE2 transfected mice and that and there were no significant differences in the neutralization abilities of the serum collected at either time point (**Fig 3C** and **S2B Fig**).

As the GFP and hACE2 transfected mice generated a robust anti-RBD IgG antibody and SARS-CoV-2 neutralizing responses following infection, we wanted to determine if this model recapitulated the association of RBD binding and serum neutralization potential that has been reported in humans [10,11]. To address this, we completed linear regression and Pearson correlation analysis comparing the RBD binding area under the curve (AUC) values to the 1/$FRNT_{50}$ neutralization values from both groups of mice (**Fig 3D**). Indeed, there was a strong correlation between SARS-CoV-2 RBD binding and neutralization potential of murine polyclonal sera from both the GFP and hACE2 transfected SARS-CoV-2 infected Ifnar1$^{-/-}$ mice. These results are consistent with reports in infected humans, [10,11], and support the use of the murine model to study the neutralizing antibody response to SARS-CoV-2.

To demonstrate that SARS-CoV-2 infection of Ifnar1$^{-/-}$ mice induced a potent high affinity antibody response independent of hACE2 expression, we also looked for differences in the T follicular helper cells ($T_{fh}$) at 5 days post SARS-CoV-2 boost. $T_{fh}$ cells are a subset of CD4+ T cells that aid antigen-specific B cells in affinity maturation and the development of a high affinity antibody response [46]. As expected from the neutralization and RBD binding data, we did see elevated frequencies of $T_{fh}$ cells in the spleens both groups of infected mice, relative to a naïve control (**S2C Fig**). There were no differences noted in the frequency of $T_{fh}$ cells between the GFP and hACE2 transfected SARS-CoV-2 infected Ifnar1$^{-/-}$ mice. From these data, we concluded that Ifnar1$^{-/-}$ mice infected with SARS-CoV-2 can generate a high affinity antibody response to the virus and this response is independent of the hACE2 expression.

Using the prime boost protocol outlined in **Fig 3A**, we also examined the CD8$^+$ and CD4$^+$ T cells from the blood after primary infection, and boost, by intracellular cytokine staining for the detection of interferon gamma (IFN-γ) in response to anti-CD3 stimulation (**Fig 3E** and **S3 Fig**). During the acute infection the IFN-γ production in both the CD4$^+$ and CD8$^+$ T cells isolated from both the GFP and hACE2 transfected group were slightly above the background of unstimulated cells isolated from the same group of mice and were not significantly different from one another. However, following the second infection with SARS-CoV-2, the IFN-γ response was significantly higher in the hACE2 transfected mice (average CD8$^+$ response = 1.5% and average CD4$^+$ response = 1.6%) as compared to GFP (average CD8$^+$ response = 0.5% and average CD4$^+$ response = 0.4%) (p = 0.025 and 0.0009, respectively). Importantly, we did not see a significant increase in the CD8$^+$ or CD4$^+$ IFN-γ T cell responses in the boosted GFP mice as compared to the GFP mice following primary infection. This was in contrast to the boosted response we observed upon secondary infection of the hACE2 transfected mice. The presence of an anamnestic T cell response upon boost suggests that both a SARS-CoV-2 specific CD8$^+$ and CD4$^+$ T cell response was primed only in the hACE-2 mRNA transfected mice.

The results of these studies indicate that the expression of hACE2 on the murine cells allows SARS-CoV-2 to enter the murine cells and undergo viral replication. This viral replication provides the antigen processing and presentation pathways access to the SARS-CoV-2 proteins. The proteins can then be either directly or cross-presented to T cells for the development of a robust CD8+ and CD4+ T cell response in comparison to the mice lacking hACE2 expression (GFP). This result is supported both by *in vitro* studies in the 3T3 cells (**Fig 1D**) and the *in vivo* T cell and FRNT studies (**Fig 3**). Combining these studies suggests that hACE2 expression

delivered by the mRNA construct allows virus entry and replication which is required for the production of viral epitopes needed to drive a T cell response.

## Viral replication

To test whether mRNA expression of hACE2 would allow the detection of infectious virus output, we followed a similar protocol to the one we had used in the adaptive immune studies where we administered 10 μg of the hACE2 mRNA or control GFP mRNA to Ifnar1$^{-/-}$ mice one day prior to infection. Twenty-four hours post transfection, the Ifnar1$^{-/-}$ mice receiving either the hACE2 or the GFP mRNA constructs were infected with 5x10$^4$ FFU per mouse using a combined IV and IN route. Three days post SARS-CoV-2 infection, the mice were harvested and viral titers were measured in the brain, kidney, spleen, liver, lung and whole blood both by qRT-PCR and by focus forming assay (FFA). We were able to detect viral genome copies in the transfected Ifnar1$^{-/-}$ mice in each organ, which were not present in SARS-CoV-2 naïve organs (**S4A–S4F Fig**). However, we did not detect differences in viral genome copies between the hACE2 and the GFP transfected mice. Additionally, we did not detect infectious virus above the limit of detection in any organ from either group of mice by FFA. These results suggest that mRNA expression of hACE2 did not enhance SARS -CoV-2 infection of the Ifnar1$^{-/-}$ mice by this dose and route.

## SARS-CoV-2 CD8$^+$ and CD4$^+$ T cell epitope identification

While the mRNA delivery of hACE2 followed by infection with 5x10$^4$ FFU of SARS-CoV-2 did not allow our group to study viral pathogenesis in Ifnar1$^{-/-}$ mice, the delivery of hACE2 mRNA to our murine model allowed for the induction of a strong CD4$^+$ and CD8$^+$ T cell response to SARS-CoV-2 (**Fig 3E**). We chose then to use our SARS-CoV-2 murine model to identify the SARS-CoV-2-specific CD4$^+$ and CD8$^+$ T cell responses using a peptide library screening method (**Figs 4 and 5**). Our group has previously successfully employed this approach for identifying Zika virus (ZIKV) T cell epitopes using a ZIKV peptide library in C57BL/6 mice [37,47]. To identify T cell epitopes for SARS-CoV-2, peptide libraries spanning the structural genes including the spike (BEI: NR-2669), nucleocapsid (N) (BEI: NR-52404), envelope (Env) (BEI:NR-52405) and Membrane (Mem) (BEI: NR-52403) were screened. Each of the peptide arrays of 12-mer to 20-mers overlapped by approximately 10 amino acids and spanned the entire length of each protein. For the initial peptide screening, a SARS-CoV Urbani strain (GenBank: AY278741) Spike peptide library (BEI: NR-2669) was used in place of SARS-CoV-2 Spike (BEI: NR-52402) due to lack of reagent availability. For the Env, NC, and Mem screening assays, peptide libraries generated from the sequences of SARS-CoV-2 USA-WA1/2020 strain were used. Each peptide from the library was spread across five plates with a total of 269 coronavirus structural peptides, that were screened in the initial studies.

To identify the epitope targets of the SARS-CoV-2 specific CD8$^+$ and CD4$^+$ T cells in the primary screen, a similar prime-boost infection strategy detailed in **Fig 3A** was followed, where hACE2 or GFP mRNA was administered, followed by infection with 5x10$^4$ FFU of SARS-CoV-2. After 29 days, the mice were again transfected with hACE2 mRNA followed 24 hours later with 5x10$^4$ FFU of SARS-CoV-2. At day 5 post SARS-CoV-2 boost, mice were sacrificed and spleens were homogenized into single cell suspensions. The splenocytes were stimulated in the presence of Brefeldin A (BFA) and a gene specific peptide pool (**Fig 4A**) or pool of peptides from the same well of the 96-well library plates (**Fig 4B**). After stimulation, splenocytes were stained with the cell surface antibodies, α-CD8, α-CD4, and α-CD19, then stained intracellularly with antibodies against the mouse cytokines interferon gamma (IFN-γ) and tumor necrosis factor-α (TNF-α) as shown in **S3 Fig**. To identify antigen experienced CD8$^+$ T

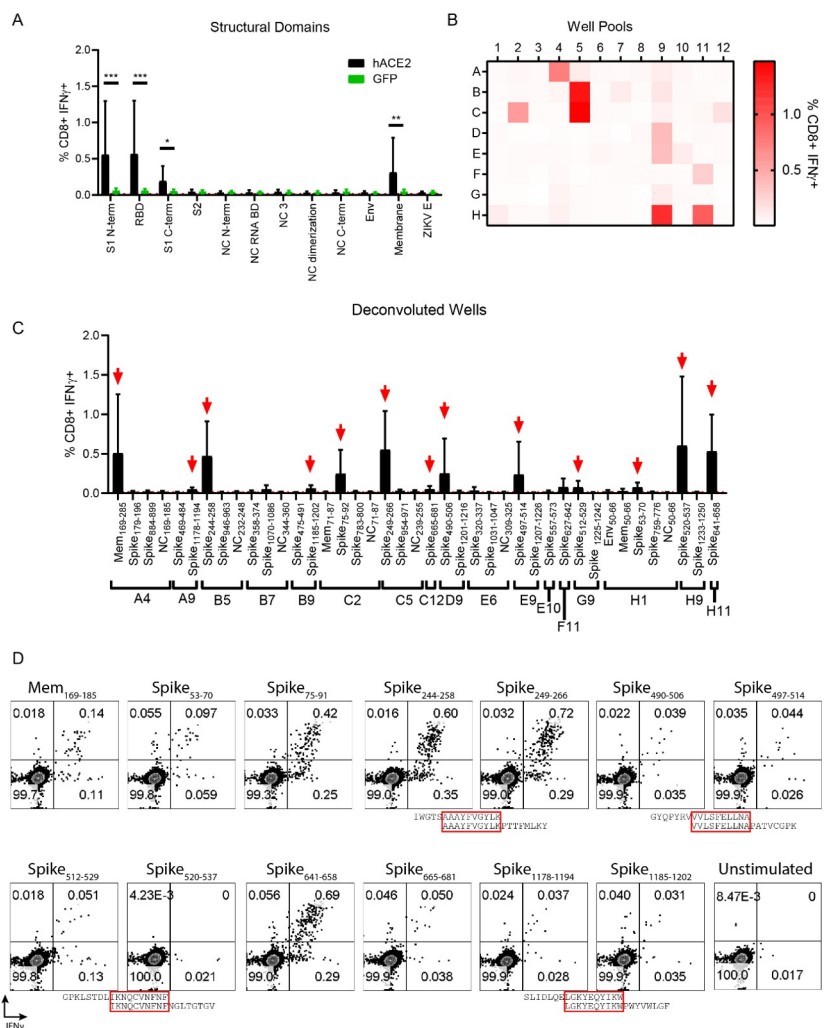

**Fig 4. In vivo transfection of hACE2 mRNA permits the detection and functional mapping of SARS-CoV-2 specific CD8+ T cell responses.** (**A**) CD8+ T cell responses to pooled peptide domains. Each equimolar peptide library pools was demarcated by peptides contained in functional domains of each protein (11 total pools). Five days post boosted infection with SARS-CoV-2 following mRNA transfection harvested splenocytes were stimulated with each peptide pool. Cells were stained to evaluate the frequency of responsive CD8+ T cells by IFN-γ expression. (**B**) CD8+ T cell responses to smaller well peptide pools. Each library was incorporated into multiple 96-well plate formats (**S5 Fig**). Within the same layout, wells from the plates were pooled such that all A1 peptides were pooled, all A2 peptides, etc. maintaining the 96-well plate format reducing the overall number of screened samples. As in 3A splenocytes boosted mice were harvested and stimulated with each peptide pool. The frequency of IFN-γ⁺ CD8⁺T cells—the magnitude of which represents responsiveness to a peptide in the pool, is enumerated in a heat map format as the average responses of 3 mice. (**C**) Identified potential well hits, were deconvoluted and used individually to stimulate splenocytes from hACE2 transfected, SARS-CoV-2 infected mice. 13 potential epitopes were identified (marked with red arrows) as defined by the frequency of IFN-γ⁺ CD8⁺T cells being at least 2-fold above background (stimulated with vehicle) in at least 3 of the 4 mice screened. (**D**) Representative flow cytometry plots displaying IFN-γ and TNFα expression in CD8 + T cells for each putative epitope in comparison to a vehicle control. Due to the overlapping nature of the peptide library, Spike$_{244-258}$ and Spike$_{249-266}$, likely are demonstrating responsiveness to the same peptide epitope. 3 other instances of this phenomenon are denoted with red boxes surrounding the amino acid sequence overlap. Statistical significance was determined by Mann-Whitney test (*p = .0159, **p = 0.0079).

cells from the SARS-CoV-2 boosted animals, we compared the results of the peptide stimulation against unstimulated cells and a ZIKV envelope peptide pool as negative controls, and anti-CD3 (clone 45-2C11) stimulated cells as a positive control. Pools that demonstrated an

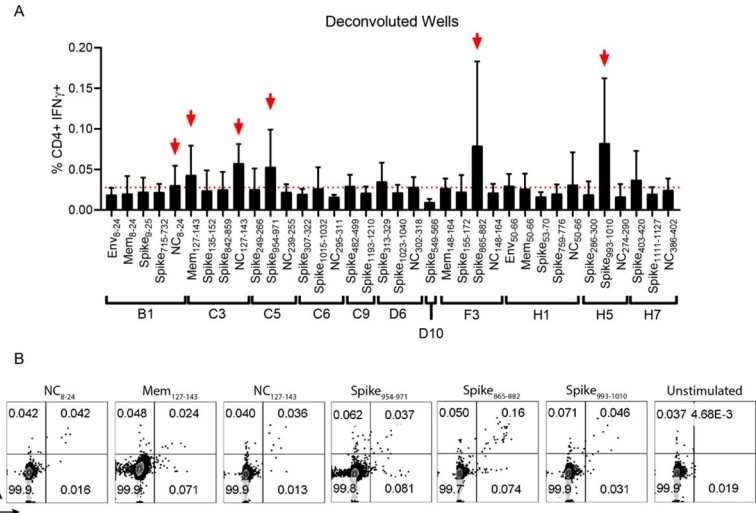

**Fig 5. In vivo transfection of hACE2 mRNA permits the detection and functional mapping of SARS-CoV-2 specific CD4⁺ T cell responses.** (**A**) As potential "well" hits were identified, the peptides contained in each well were deconvoluted and used individually to stimulate splenocytes from hACE2 transfected, SARS-CoV-2 infected mice. 6 potential epitopes were identified (marked with red arrows) as defined by the frequency of IFN-γ⁺ CD4⁺ T cells being at least 2-fold above background (stimulated with vehicle) in at least 3 of the 4 mice screened. (**B**) Representative flow cytometry plots displaying IFN-γ and TNFα expression in CD4⁺ T cells for each putative epitope in comparison to a vehicle control.

IFN-γ response that was more than two-fold over background in the majority of mice screened were brought forward to identify possible SAR-CoV-2 epitopes.

Using this approach, it became clear that the majority of the CD8+ T cell response in these H2ᵇ restricted mice targeted peptides within the spike and Mem proteins (**Fig 4A**). To further narrow down potential epitope targets, peptides were pooled by combining the same well from each of the five plates from the aliquoted library shown in **S5 Fig** and **S1 Table.** These pools were used to stimulate splenocytes of hACE2 mRNA transfected and SARS-CoV-2 boosted mice (**Fig 4B**). Using this approach 17 well pools screened positive for potential SARS-CoV-2-specific CD8⁺ T cell epitopes. The peptide pools were deconvoluted and individual multimer peptides from each presumptive positive well was evaluated (**Fig 4C**). The individual peptides within the peptide pools were screened by repeating the hACE2 transfection and SARS-CoV-2 boosting strategy in the Ifnar1⁻ᐟ⁻ mice. With this approach, we identified 13 peptide hits that induced a cytokine response that was greater than 2-fold above background in the majority of mice screened (**Table 1**). We named the SARS-CoV-2 peptide peptides with the abbreviated name of the viral protein followed by the number of the amino acid based upon the SARS-CoV-2 open reading frame for each gene, for example Mem₁₆₉₋₁₈₅ would mean the epitope began at the 169th amino acid in the Mem open reading frame. To demonstrate that the responses to each peptide were polyfunctional, we assessed IFN-γ and TNF-α expression (**Fig 4D**). All 13 of the peptides we screened were polyfunctional based upon co-expression of both IFN-γ and TNF-α in response to stimulation.

The identical approach, using the same mice, was concurrently used to characterize the CD4⁺ T cell responses to SARS-CoV-2 (**Fig 5 and S6 Fig**). Stimulation of the CD4⁺ T cells with the gene specific peptide pool and the pool peptides from the combined five plates showed weak but detectable responses spread across all 12 protein domains evaluated (**S6A Fig**). While the signal to noise ratio of the CD4⁺ T cell response was lower than what we had observed for the CD8⁺ T cells, the responses from the wells of pooled peptides from the protein

domains suggested that the strongest CD4$^+$ T cell responses were located in the S2 region of the spike protein and the NC RNA binding domain. By evaluating the smaller well pools of peptides consisting of 1–5 pooled peptides, we identified 11 positive well pools, where the IFN-γ responses to the peptide pool was greater than 2-fold over background in multiple animals (S6B Fig). By deconvoluting those wells to individual peptides, we identified six novel SARS-CoV-2 reactive CD4$^+$ T cell epitopes in C57BL/6 mice (Fig 5A and Table 2). Of the six CD4$^+$ T cell epitopes identified one was in Mem, two were in NC, and three were in spike. Similar to what we observed for the peptide stimulated CD8$^+$ T cells, the CD4$^+$ T cells stimulated with the individual peptide epitopes were able to make both IFN-γ and TNF-α in response to peptide stimulation (Fig 5B).

## Identification of optimal 8-mer and 9-mer H2$^b$ restricted SARS-CoV-2 CD8$^+$ T cell epitopes

Based upon our functional SARS-CoV-2 library-based epitope mapping, we identified 13, 12-18-mer peptides which contained H2$^b$ restricted CD8$^+$ T cell epitopes (Table 2). However, these findings were limited for 2 reasons: 1) The majority of the hits were identified in the Spike protein, for which the SARS-CoV peptide library was used and 2) CD8$^+$ T cells primarily respond optimally to 8-mer or 9-mer peptides presented in the context of MHC and these library peptides used for screening contained 12-18-mer peptides. These 12-18-mer SARS-CoV peptides undoubtedly stimulated a T cell response *ex vivo*, but the optimal SARS-CoV-2 peptide sequences were still not confirmed. Therefore, we sought to determine the optimal 8-mer or 9-mer H2$^b$ restricted SARS-CoV-2 CD8+ T cell peptide epitopes. We analyzed the analogous SARS-CoV-2 multimer hit sequences (Table 1) and based upon known K$^b$ and D$^b$ conserved peptide anchor residues [48], generated a list of potential optimal SARS-CoV-2 8-mer or 9-mer peptides for each library hit and purchased these peptides (Table 3). Based upon these sequences' predicted binding affinities, it appeared that all of these epitopes were likely K$^b$ restricted.

To test the response to each of these epitopes in H2$^b$ restricted SARS-CoV-2 infected mice, we repeated the same priming and boosting procedure outlined in Fig 3A and harvested

**Table 1. Functional identification of SARS-CoV-2 CD8$^+$ T cell epitopes.** [1]Peptide name is based on the protein they are contained within, followed by the number of the first to the last amino acid residue of the peptide in the context of the full protein. Sequences contained within the "spike" peptide library correspond with SARS-CoV spike. [2]Exact amino acid residues of peptide used to stimulate splenocytes. [3] Average fold over background IFN-γ$^+$ CD8$^+$T cells stimulated with listed peptides. Background is defined as the frequency of IFN-γ$^+$ CD8$^+$T cells in a well stimulated with vehicle control. The average and standard deviation are from three independent experiments.

| CD8$^+$ Library Multimer Hit | Peptide Name[1] | | Peptide Library Sequence[2] | Fold Over Background[3] | |
|---|---|---|---|---|---|
| | | | | Avg. | Std. Dev |
| 1 | SARS-CoV-2 | Mem$_{169-185}$ | TVATSRTLSYYKLGASQ | 16.2 | +/-9.3 |
| 2 | SARS-CoV | Spike$_{53-70}$ | YLTQDLFLPFYSNVTGFH | 4.4 | +/-0.7 |
| 3 | SARS-CoV | Spike$_{75-91}$ | TFGNPVIPFKDGIYFAA | 8.4 | +/-4.3 |
| 4 | SARS-CoV | Spike$_{244-258}$ | IWGTSAAAYFVGYLK | 31.6 | +/-5.4 |
| 5 | SARS-CoV | Spike$_{249-266}$ | AAAYFVGYLKPTTFMLKY | 39.3 | +/-6.9 |
| 6 | SARS-CoV | Spike$_{490-506}$ | GYQPYRVVVLSFELLNA | 2.2 | +/-0.2 |
| 7 | SARS-CoV | Spike$_{497-514}$ | VVLSFELLNAPATVCGPK | 2.8 | +/-1.1 |
| 8 | SARS-CoV | Spike$_{512-529}$ | GPKLSTDLIKNQCVNFNF | 4.5 | +/-1.9 |
| 9 | SARS-CoV | Spike$_{520-537}$ | IKNQCVNFNFNGLTGTGV | 71.3 | +/-22.7 |
| 10 | SARS-CoV | Spike$_{641-658}$ | HVDTSYECDIPIGAGICA | 41.1 | +/-7.5 |
| 11 | SARS-CoV | Spike$_{665-681}$ | LLRSTSQKSIVAYTMSL | 4.0 | +/-0.7 |
| 12 | SARS-CoV | Spike$_{1178-1194}$ | SLIDLQELGKYEQYIKW | 5.0 | +/-1.5 |
| 13 | SARS-CoV | Spike$_{1185-1202}$ | LGKYEQYIKWPWYVWLGF | 7.0 | +/-4.7 |

**Table 2. Functional identification of SARS-CoV-2 CD4+ T cell epitopes.** [1]Screened peptide sequences are named based on the protein they are contained within, followed by the number of the first amino acid residue of the peptide in the context of the full protein to the last amino acid residue. Peptides within the "Spike" peptide library correspond with SARS-CoV Spike due to library availability. [2] Average fold over background IFN-$\gamma^+$ CD4$^+$T cells stimulated with listed peptides. Background is defined as the frequency of IFN-$\gamma^+$ CD4$^+$T cells in a well stimulated with vehicle control. The average and standard deviation is from three independent experiments [3]Analogous SARS-CoV-2 peptide name in instances where SARS-CoV peptide library had to be used due to reagent availability. [4]SARS-CoV-2 peptide sequence in instances where SARS-CoV peptide library was used the analogous SARS-CoV-2 peptide sequence is noted.

| Peptide Screened[1] | | Fold Over Background[2] | | SARS-CoV-2 Peptide[3] | SARS-CoV-2 Sequence[4] |
|---|---|---|---|---|---|
| | | Avg. | Std. Dev | | |
| SARS CoV-2 | Mem$_{127-143}$ | 2.7 | +/-1.5 | Mem$_{127-143}$ | TILTRPLLESELVIGAV |
| SARS CoV-2 | NC$_{127-143}$ | 5.0 | +/-4.0 | NC$_{127-143}$ | KDGIIWVATEGALNTPK |
| SARS CoV-2 | NC$_{8-24}$ | 3.6 | +/-3.3 | NC$_{8-24}$ | NQRNAPRITFGGPSDST |
| SARS CoV | Spike$_{865-882}$ | 4.5 | +/-3.9 | Spike$_{883-900}$ | TSGWTFGAGAALQIPFAM |
| SARS CoV | Spike$_{954-971}$ | 3.2 | +/-2.0 | Spike$_{972-989}$ | AISSVLNDILSRLDKVEA |
| SARS CoV | Spike$_{993-1010}$ | 4.8 | +/-2.9 | Spike$_{1011-1028}$ | QLIRAAEIRASANLAATK |

**Table 3. Optimal 8-mer and 9-mer SARS-CoV-2 epitope identification.** Based on the analogous SARS-CoV-2 peptide library sequences. Peptide sequences are named based on the protein they are contained within, followed by the number of the first amino acid residue of the peptide in the context of the full protein, to the last amino acid residue. potential optimal 8-mer or 9-mer CD8+ T cell epitopes were predicted. To determine the predicted binding affinities of the putative SARS-CoV-2 8-mer or 9-mer CD8+ T cell peptide epitopes to MHC class I, using the NetMHCpan 3.0 server, which uses artificial neural networks to predict relative binding affinities of peptides to any Kb or Db. As controls for strong Kb and Db binders respectively, Ova peptide and ZIKV E294 were input into the same algorithm. Optimal peptide epitopes are highlighted based on functional T cell data in combination with RMA-S stabilization assay data.

| Library Multimer Hit Number | Peptide name | SARS-CoV-2 8-mer or 9-mer AA sequence | Predicted Binding Affinities ($K_d$) | |
|---|---|---|---|---|
| | | | K$^b$ (nM) | D$^b$ (nM) |
| 1 | Mem$_{176-183}$ | LSYYKLGA | 507.8 | 40626.6 |
| | Mem$_{177-185}$ | SYYKLGASQ | 6898.4 | 35593.9 |
| | Mem$_{174-182}$ | RTLSYYKLG | 2599.6 | 38590.6 |
| | **Mem$_{174-181}$** | **RTLSYYKL** | **31.4** | 27304.9 |
| 2 | Spike$_{54-62}$ | LFLPFFSNV | 140.5 | 31735.7 |
| | **Spike$_{55-62}$** | **FLPFFSNV** | **257.7** | 30022.8 |
| | Spike$_{51-59}$ | TQDLFLPFF | 3484.6 | 20441.6 |
| 3 | **Spike$_{77-85}$** | **KRFDNPVLP** | **31628.7** | 33276.6 |
| | Spike$_{77-84}$ | KRFDNPVL | 10291.3 | 30062.1 |
| 4 and 5 | **Spike$_{263-270}$** | **AAYVGYL** | **25.8** | 15164.7 |
| | Spike$_{264-272}$ | AYYVGYLQP | 4732.7 | 34345 |
| 6 and 7 | Spike$_{511-518}$ | VVLSFELL | 15.3 | 21549.5 |
| | Spike$_{511-519}$ | VVLSFELLH | 15356.6 | 41896.2 |
| | **Spike$_{513-520}$** | **LSFELLHA** | **1392.2** | 31424.4 |
| 8 and 9 | **Spike$_{539-546}$** | **VNFNFNGL** | **3.7** | 19584.5 |
| | Spike$_{539-547}$ | VNFNFNGLT | 245.7 | 36271.9 |
| 10 | **Spike$_{656-664}$** | **VNNSYECDI** | **4668** | 33602.9 |
| | Spike$_{656-663}$ | VNNSYECD | 32944.1 | 47773.5 |
| | Spike$_{658-665}$ | NSYECDIP | 25287.3 | 39875 |
| 11 | Spike$_{692-699}$ | IIAYTMSL | 121.6 | 34624.1 |
| | **Spike$_{691-699}$** | **SIIAYTMSL** | **34** | 8104.1 |
| 12 and 13 | **Spike$_{1204-1212}$** | **GKYEQYIKW** | **15996.1** | 38545.1 |
| | Spike$_{1205-1213}$ | KYEQYIKWP | 34608 | 46279.8 |
| Kb control | Ova | SIINFEKL | 44.4 | 3514.6 |
| Db control | ZIKV E$_{294}$ | IGVSNRDFV | 7456.2 | 25.8 |

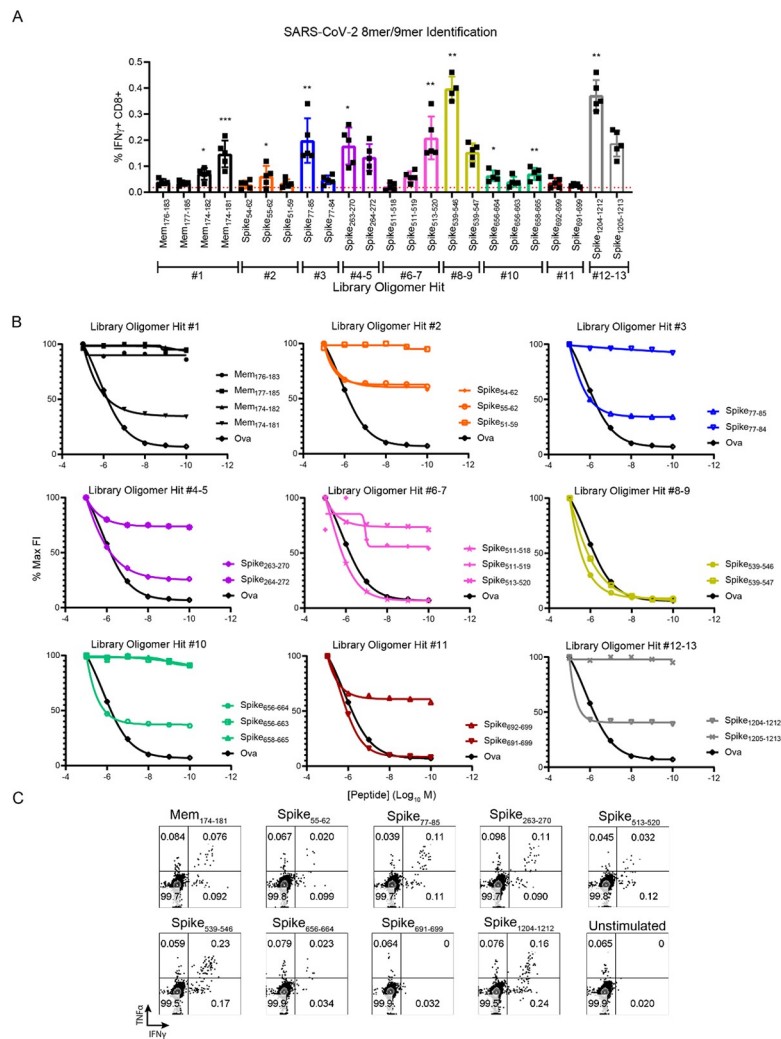

**Fig 6. Identification of optimal 8-mer and 9-mer peptide epitopes.** (**A**) T cell response to 8-mer or 9-mer peptides. Multiple SARS-CoV-2 8-mer or 9-mer peptide variants from each potential library hit were identified and purchased based on known $K^b$ or $D^b$ anchor residues. Five days post boosted infection with SARS-CoV-2 following transfection with hACE2 mRNA, splenocytes were harvested and stimulated for 6 hours with each peptide in the presence of brefeldin A. After stimulation, cells were stained for flow cytometry to evaluate the frequency of responsive CD8+T cells by IFN-γ expression. Each color is indicative of a peptide variant being derived an individual potential library hit. (**B**) $K^b$ RMA-S stabilization assay. To determine relative ability of individual peptide variants to stabilize the $K^b$ molecule, decreasing concentrations of each peptide variant were incubated for 4 hours with TAP deficient RMA-S cells at 29 degrees C before being moved to 37 degrees C for 1 hour. Cells were then stained for either $K^b$ or $D^b$ with fluorescently conjugated antibodies and geometric mean fluorescent intensity (gMFI) was measured using an Atttune focusing flow cytometer. Fluorescence index (FI) was determined by dividing the gMFI of cells pulsed with peptide by cells with no peptide. Data is presented as a percentage of the maximum FI for each peptide. As a positive control, the $K^b$ restricted peptide Ovalbumin (SIINFEKL) was used. (**C**) Representative cytokine responses to each optimal epitope of mice transfected with hACE2 and infected with SARS-CoV-2 five days post boost.

splenocytes five days after SARS-CoV-2 boost. The response to each potential optimal epitope was determined by IFN-γ production by CD8+ T cells compared to cells stimulated with vehicle (**Fig 6A**). We were able to detect responses above background in all 8-mer or 9-mer variants evaluated. Interestingly, in some cases multiple variants of the same core library sequence had statistically significant CD8+ T cell responses compared to vehicle stimulated wells, suggesting that in some cases more than one peptide variant could be used to achieve T cell

activation as measured by cytokine production. Though for the generation of MHC tetramers, it is ideal to determine which peptide variant best stabilizes class I MHC.

To further elucidate which 8-mer or 9-mer variants represented optimal peptides, we performed RMA-S stabilization assays which measures a peptide's intrinsic ability to stabilize either $K^b$ or $D^b$ through the use of a TAP deficient cell line which lacks the ability to process endogenous antigens. An optimal peptide should manifest as a high fluorescence index when considering staining for $K^b$ or $D^b$, which decreases in a dose dependent manner. As expected based on the peptide sequences and predicted binding affinities, no peptide variants were $D^b$ restricted (**S7 Fig**). However, when evaluating a peptide's ability to stabilize $K^b$ compared to a known $K^b$ peptide Ova, it became clear in every case that a single 8-mer or 9-mer peptide of each group of variants was the optimal peptide (**Fig 6B**). Based upon 2 factors: 1) The magnitude of the $CD8^+$ T cell response to the peptide variant and 2) The ability of the variant to stabilize $K^b$ we determined the optimal 8-mer or 9-mer peptide sequences of 9 $H2^b$ restricted SARS-CoV-2 specific $CD8^+$ T cell epitopes: $Mem_{174-181}$, $Spike_{55-62}$, $Spike_{77-85}$, $Spike_{263-270}$, $Spike_{513-520}$, $Spike_{539-546}$, $Spike_{656-664}$, $Spike_{691-699}$, and $Spike_{1204-1212}$ (**Fig 6A and 6B and Table 3**). Most of these 9 optimal epitopes stimulated polyfunctional T cell responses, as measured by IFN-γ and TNF-α production by $CD8^+$ T cells (**Fig 6C**). Important for the analysis of RBD subunit vaccines, SARS-CoV-2 specific $CD8^+$ T cell epitope $Spike_{513-520}$ is located within the RBD (**S8 Fig**), which will allow for the analysis of both T cell and antibody responses to these vaccine constructs. Finally, we confirmed responses against these optimal epitopes in SARS-CoV-2 infected K18-hACE2 transgenic mice (**S9 Fig**). These mice express the hACE2 receptor under the expression of the keratin 18 promotor, directing expression to epithelial cells and are a current model of SARS-CoV-2 infection and disease [32]. T cell responses to each optimal 9/8-mer were detectable in the K18-hACE2 transgenic mice. Collectively, these data demonstrate specific metrics for the evaluation of SARS-CoV-2 specific T cell responses, which will be important for defining correlates of protection and evaluating vaccine efficacy.

## Discussion

There is a significant gap in our knowledge concerning infection and correlates of protection for coronaviruses, including the novel coronavirus SARS-CoV-2. Much of that deficit is due to the lack of tractable small animal models in which the reagents necessary to study T and B cell responses are available. The goal of this study was to use an mRNA delivery system to express hACE2, the putative receptor for SARS CoV-2 in mice, to generate a small animal model to study the adaptive immune response to SARS CoV-2 infection.

By developing a highly tractable hACE2 expression system, we have generated the potential for an animal model which can be rapidly used by multiple groups to study the T cell response to SARS-CoV-2. In most cases the ideal murine model is both susceptible to pathogen infection and has some degree of immunodeficiency. For the hACE2 transgenic mice the crossing of the transgenic mice onto a knockout background is both time consuming and expensive. Methods of hACE2 expression including AAVs and adenoviral vectors are limited by cell and species tropism. mRNA expression of hACE2 removes these barriers allowing for hACE2 expression in multiple species using a single construct. Additionally, mRNAs can be targeted to specific organs including the liver and lungs [49,50], allowing for localized expression of hACE2 to study the impact of SARS-CoV-2 on specific organs. Finally, as we and others have shown, the mRNA transfection is poorly immunogenic, so repeated administration of the same or different constructs can be administered to the same animal over a prolonged period of time. As we and others have shown, flexibility and simplicity of the mRNA transfection system make it an ideal method for use in the study of the immune response to SARS-CoV-2.

Previous studies have shown that administration of higher doses of virus by the intranasal route did lead to the detection of virus in mice which had been induced to express hACE2 through the administration of an adeno-associated virus vector or recombinant adenovirus [2,8,22–28]. So, while hACE2 mRNA transfection into the Ifnar1[-/-] mice did not measurably increase the susceptibility of the Ifnar1[-/-] mice to SARS-CoV-2, it is possible that the dose and route of infection as well as the harvest time points we chose for these studies were not optimal for detection of infectious virus. In the current study, we chose low dose administration of virus to focus on immune responses with the plan in future studies, to use a higher doses of virus, similar to those published in other mouse models [2,8,22–28] to determine the utility of the mRNA transfection system for SARS-CoV-2 pathogenesis. Using this low dose strategy, we were able to detect a potent neutralizing antibody response in mice that was directed against RBD, similar to what has been seen in humans [10,11]. While this response was detected independent of hACE2 transfection, the presence of RBD-specific neutralizing antibodies in mice further supports the use of mice as a viable preclinical animal model for the study of vaccines against SARS-CoV-2. Additionally, future studies with alternate routes will be useful in determining if the delivery of mRNA expressing hACE2 can be used to develop a mouse model to study virus pathogenesis in specific tissues or the impact of human ACE2 variants on disease.

Understanding the role of the antigen specific antibody and T cell responses against SARS-CoV-2 is critical for the development of safe and effective vaccines. Studies of SARS-CoV and Middle East respiratory syndrome coronavirus (MERS-CoV) suggest that the RBD of the spike glycoprotein is likely the most effective target for current antibody-based vaccines against SARS-CoV-2. However, these SARS-CoV and MERS-CoV vaccine studies also noted that there is a concern that a vaccine directed solely against the spike glycoprotein will induce viral escape [10,11], and a successful vaccine should contain both a strong T cell response for early viral control, and neutralizing antibody for viral clearance [51–53]. Additional studies with murine hepatitis virus (MHV) A59, a natural mouse coronavirus pathogen [54], concluded that a T cell response in combination with an antibody response was critical for controlling a coronavirus infection [55]. Therefore, understanding the role of the T cell as well as the antibody response against SARS-CoV-2 will be critical for the development, testing and evaluation of future vaccines.

The reference strain of SARS-CoV-2 shares approximately 82% identity with the SARS-CoV reference strain (NC_004718.3) at the nucleotide level, with the greatest nucleotide differences occurring in the replicase, spike, and an accessory gene, 8a (GISAID, [56]). Interestingly, comparison at the amino acid level shows an approximate 77% identity, with the divergence of amino acids not localized to a particular protein shared between the two viruses. The similarities between SARS-CoV and SARS-CoV-2 have led to evidence of considerable serological cross-reactivity [57]. As clinical isolates of SARS-CoV-2 have been sequenced, the genomic data published on GISAID and NCBI support a high degree of conservation at the amino acid level between the clinical isolates, particularly within the spike protein of SARS-CoV-2. In humans, T cell epitope analysis suggests that the majority of the CD4[+] and CD8[+] T cell responses identified in SARS-CoV-2 positive patients are directed toward spike with responses also detected against Mem and NC [3,4]. This combination of a focused immune response toward structural proteins and minimal degree of sequence variation within the spike protein suggests that vaccines that induce strong responses to the SARS-CoV-2 spike protein will induce T cell as well as antibody responses to the virus.

We have identified CD4[+] and CD8[+] SARS-CoV-2 specific T cell epitopes derived from spike, NC and Mem. We reported both the amino acid residues of the SARS-CoV peptide sequences that were screened and the analogous amino acid residues for the SARS-CoV-2

peptide sequences for the Spike protein. In the case of the CD4+ T cell epitopes, unlike what we had seen for the CD8+ T cell epitopes, we noted only one amino acid difference; a change from an A to S between the SARS-CoV Spike$_{865-882}$ and SARS-CoV-2 Spike$_{883-900}$. For the identified CD4+ T cell epitopes we saw no other changes in the amino acid sequences between the SARS-CoV and SARS-CoV-2 spike epitopes identified. For this reason, we chose not to synthesize the highly similar, and often identical SARS-CoV-2 peptide sequences, for confirmation of the CD4+ T cell epitopes. Due to our decision to screen with the SARS-CoV peptide library for Spike, there is a possibility that there are SARS-CoV-2 non-cross-reactive CD4+ and CD8+ T cell epitopes that were not picked up by the screen. Future screens with a Spike peptide library generated from SARS-CoV-2 will be required to resolve this issue.

In the present study, we utilized this system to understand the adaptive immune response during SARS-CoV-2 infection in Ifnar1$^{-/-}$ mice. We were able to demonstrate that multiple administrations of hACE2-encoding mRNA can be used to detect enhanced CD4+ and CD8+ T cell responses to the structural proteins of SARS-CoV-2 during a prime and boost infection (**Fig 3**). Additionally, by using this approach in conjunction with an overlapping peptide library to stimulate these T cells, we identified nine SARS-CoV-2 CD8+ T cell epitopes and six CD4+ T cell epitopes which are H2$^b$ restricted (**Figs 4–6**). The identification of the SARS-CoV-2 specific CD4+ and CD8+ T cell epitopes is an essential first step in establishing the murine preclinical model for vaccine and therapeutic evaluations. Importantly as was seen in the human studies [3,4], the SARS-CoV-2-specific CD4+ and CD8+ T cell epitopes we identified in the mouse also target spike, NC and Mem. Our results expand the knowledge within the field of the adaptive immune response to SARS-CoV-2, moving much-needed pre-clinical animal models for SARS-CoV-2 and COVID-19 forward.

## Materials and methods

### Ethics statement

All animal studies were conducted in accordance with the Guide for Care and Use of Laboratory Animals of the National Institutes of Health and approved by the Saint Louis University Animal Care and Use Committee (IACUC; protocol 2771).

### Generation of hACE2 and GFP constructs

GFP and hACE2 expression constructs were cloned by Gibson assembly into a pUC57 vector (sequences available). The plasmids were linearized downstream of the 3' UTR and polyA tail by XbaI digestion. The linearized plasmids were purified and utilized as templates in a T7 ARCA in vitro transcription reaction (New England Biolabs). The mRNA product was then purified using an Invitrogen Purelink RNA mini kit according to the manufacturer's instructions. Transcript length and quality was confirmed by RNA bleach gel.

### Virus and cells

SARS-CoV-2 (Isolate USA-WA1/2020) was obtained from BEI (catalog NR-52281) A p1 stock was grown in African green monkey kidney cells (Vero- E6) purchased from ATCC using this initial seed stock. A p2 stock was then grown from this p1 stock by infecting Vero- E6 cells at an MOI of 0.01 in complete DMEM and harvested at 96 hours post infection. In vitro transfection and infection experiments were completed using murine fibroblast 3T3 cells cultured in complete DMEM and were obtained from American Type Culture Collection (ATCC CRL-1658).

## In vitro transfection

3T3 cells were transfected with either hACE2 or GFP mRNA using an Amaxa cell line nucleofector L kit (Lonza; catalog number VCA-1005) according to the manufacturer's instructions. $5x10^5$ transfected cells were plated in each well of a 6-well dish. hACE2 mRNA stability was measured by qRT-PCR at 48, 72, and 96 hours post transfection. Stability of GFP expression was confirmed by flow cytometry at 24, 48, and 72 hours post transfection. Twenty-four hours post transfection, susceptibility of the cells was evaluated by infecting with SARS-CoV-2 at an MOI of 0.01 in 0.5 ml of DMEM. At 24, 48, and 72 hours post infection, media supernatant was removed from the wells and RNA was extracted from the cells using an Invitrogen Purelink RNA mini kit according to the manufacturer's instructions.

## Mice, in vivo transfection, and infections

Type 1 interferon receptor deficient mice (Ifnar1-/-) and K18-hACE2 transgenic mice were purchased from Jackson laboratories (stock numbers 32045 and 034860, respectively) and maintained as a colony in a pathogen-free mouse facility at Saint Louis University- School of Medicine. Eight-week-old Ifnar1-/- mice were transfected with 10 μg of RNA using Polyplus in vivo-jet RNA in vivo transfection reagent prepared according to the manufacturer's instructions and administered via intravenous (IV) and intranasal (IN) combination route (100 μl and 20 μl, respectively). For the evaluation of in vivo transfection efficiency, 24 hours post transfection, mice were administered a lethal dose of ketamine/xylazine cocktail and perfused with 20 ml of PBS. A subset of lung and liver tissue was homogenized in TriReagent RT for RNA isolation to quantify hACE2 RNA levels. A separate subset of lung and liver tissue was used to evaluate protein expression by flow cytometry. The tissues were minced into a digestion buffer containing 0.05% collagenase I, 10 μg/ml DNase, and 10 mM HEPES in HBSS. After digestion for 1 hour, cells were strained over a 100 μm cell strainer and purified using a 30% percoll gradient as previously described [37] and stained for hACE2. For infection experiments, 24 hours following transfections, mice were infected with $5x10^4$ focus forming units (FFU) of SARS-CoV-2 via IV and IN combination route (100 μl and 20 μl, respectively). A subset of mice (n = 6 GFP and n = 7 hACE2) were used to quantify viral burden at 3 days post infection. Mice were administered a lethal dose of ketamine/xylazine cocktail and perfused with 20 ml of PBS. Blood was collected into RNAsol BD, and viral RNA was extracted according to the manufacturer's instructions. Spleen, liver, kidney, brain, and lung tissues were collected into Sarstedt tubes and snap frozen. Organs were homogenized in DMEM using a bead beater and viral RNA was extracted from the organ lysates using TriReagent RT. For antibody and T cell experiments, mice were transfected with 10 μg of RNA using Polyplus in vivo-jet RNA in vivo transfection reagent prepared according to the manufacturer's instructions and administered via intravenous (IV) and intranasal (IN) combination route. 24 hours following transfections, mice were infected with $5x10^4$ focus forming units (FFU) of SARS-CoV-2 via IV and IN combination route. At day 8 post infection, blood was collected for acute phase T cell analysis. At day 10 post infection, serum was collected for SARS-CoV-2 neutralization and ELISA assays. 30 days following initial infection, mice were again transfected with RNA and infected in the same manner. At 5 days post boost, mice were humanely euthanized and splenocytes and blood were collected for epitope identification and boosted serological analysis. To confirm the identification of the CD8+ T cell epitopes found in hACE2 transfected Ifnar[1-/-] mice, in a mouse model of SARS-CoV-2 pathogenesis, K18-hACE2 transgenic mice were infected IN route with $10^4$ FFU of SARS-CoV-2. At 10 days post infection, splenocytes were harvested and stimulated with the identified peptides as described below.

## qRT-PCR

hACE2 expression was measured by qRT-PCR using Taqman primer and probe sets from IDT (assay ID Hs.PT.58.27645939). SARS-CoV-2 viral burden was measured by qRT-PCR using Taqman primer and probe sets from IDT with the following sequences: Forward 5' GAC CCC AAA ATC AGC GAA AT 3', Reverse 5' TCT GGT TAC TGC CAG TTG AAT CTG 3', Probe 5' ACC CCG CAT TAC GTT TGG TGG ACC 3'. Synthesized hACE2 RNA was used as a copy control to quantify the number of hACE2 molecules present in each sample. Similarly, a SARS-CoV-2 copy number control (available from BEI) was used to quantify SARS-CoV-2 genomes.

## T cell stimulation

For anti-CD3 stimulation of peripheral blood lymphocytes, blood was collected via submandibular cheek bleed directly into alkaline lysis buffer. After red blood cell lysis, cells were washed twice with complete RPMI media (10% FBS, 1X HEPES, and 1X beta-mercaptoethanol) and resuspended in complete RPMI. Cells were then split between 2 wells of a 96-well round bottom plate and stimulated for 6 hours at 5% $CO_2$ and 37˚C in the presence of 10 μg/ml brefeldin A with 5 μg/ml of anti-CD3 (clone 2C11) or water as a negative control. For peptide stimulation of splenocytes, spleens were harvested into complete RPMI medium from mice 5 days post SARS-CoV-2 boost. Spleens were ground over a 70μm filter and then washed with 10ml of RPMI. Approximately $5 \times 10^5$ cells were plated per well in a 96-well round bottom plate and stimulated for 6 hours at 5% $CO_2$ and 37˚C in the presence of 10 μg/ml brefeldin A and 50 μg/ml of each peptide or peptide pools. As negative controls, cells were stimulated with a pool of ZIKV envelope peptides or vehicle DMSO.

## In vivo fLuc expression

24 hours post in vivo transfection with 10 μg of fLuc encoding mRNA, mice were injected intraperitoneally (IP) with 150 μg/g of body weight of D-Luciferin potassium salt and anesthetized in an induction chamber with 4% isoflurane gas. Mice were placed in the IVIS Spectrum imaging system on the platform and maintained on 1.5% isoflurane via nose cone. 15 minutes after D-Luciferin administration, mice were imaged using a 45 second exposure time to detect luciferase activity.

## Flow cytometry

To determine in vivo transfection efficiency by flow cytometry, cells isolated from lungs and livers were stained with a human ACE2 specific primary antibody (Twist Biopharma catalog number TB-184-2) (at 1 μg per $10^6$ cells) followed by an AF-488 conjugated goat anti-human secondary antibody (1:5000) and analyzed using an Attune focusing flow cytometer. For peptide stimulation assays, following stimulation of lymphocytes, cells were washed once with PBS and stained overnight in PBS at 4˚C for the following surface antigens: CD4 (clone RM-4-5), CD8α (clone 53–6.7), and CD19 (clone 1D3). Cells were washed in PBS, then fixed in 2% paraformaldehyde at 4˚C for 10 minutes. After fixation, cells were permeabilized with 0.5% saponin and stained in 0.5% saponin at 4˚C for 1 hour for the following intracellular antigens: TNFα (clone Mab11) and IFN-γ (clone B27). After intracellular staining, cells were washed with 0.5% saponin followed by PBS. The cells were analyzed by flow cytometry using an Attune NxT focusing flow cytometer. For analysis, CD4+ T cells were gated on lymphocytes, CD19 negative, CD4 positive and CD8 negative cells. CD8+ T cells were gated on lymphocytes, CD19 negative, CD4 negative and CD8 positive (S3 Fig). Antigen specific cells were then

identified as producing IFN-γ and/or TNFα at greater than 2-fold over cells stimulated with just a vehicle control. For T follicular helper staining, the splenocytes were incubated in Fc block in PBS for 1 hour at 4 degrees Celsius. The cells were then washed with PBS and stained for CXCR5, CD62L, CD8, CD4, PD-1, CD3, B220, and CD4 before being washed with PBS and run on an Attune focusing flow cytometer. $T_{fh}$ cells were defined as lymphocytes based on forward and side scatter, singlets, B220 negative, CD3 positive, CD4 positive, PD-1 and CXCR5 high (S2 Fig).

## SARS-CoV-2 Receptor binding domain ELISA

To determine the binding potential of polyclonal sera from SARS-CoV-2 infected mice to the hACE2 receptor binding domain (RBD) of SARS-CoV-2, maxisorp ELISA plates were coated overnight at 4°C with 1 μg/ml of recombinant SARS-CoV-2 RBD protein in carbonate buffer. The following day, the plates were blocked with PBS, 5% BSA, and 0.5% Tween for 2 hours at room temperature prior to being washed. Serum from each mouse was serially diluted and added to each well and allowed to incubate for 1 hour at room temperature prior to being washed. Horseradish peroxidase conjugated goat-anti-human IgG secondary antibody was added and allowed to incubate for 1 hour at room temperature prior to being washed. TMB enhanced substrate was added and allowed to incubate in the dark at room temperature for 15 minutes prior to quenching with 1N HCl. Following quenching, absorbance of the plate was read at 450 nanometers using a BioTek Epoch plate reader.

## Focus reduction neutralization assay (FRNT)

The FRNT was completed as previously described [33]. Briefly, serum from each mouse was serially diluted in DMEM containing 5% FBS and combined with ~100 focus forming units (FFU) of SARS-CoV-2 and allowed to complex at 37°C and 5% $CO_2$ for 1 hour in a 96-well round bottom plate. The antibody-virus complex was then added to each well of a 96-well flat bottom plate containing a monolayer of Vero WHO cells. Following 1 hour of incubation at 37°C and 5% $CO_2$, the cells were overlaid with 2% methylcellulose and returned to the incubator. After 24 hours of infection, the cells were fixed with 5% electron microscopy grade paraformaldehyde in PBS for 15 minutes at room temperature. The cells adherent to the plate were then rinsed with PBS and permeabilized with 0.05% Triton-X in PBS. Foci of infected Vero cells were stained with anti-SARS polyclonal guinea pig sera (BEI) overnight at 4°C and washed 3 times with 0.05% Triton-X in PBS. Cells were then stained with horseradish peroxidase conjugated goat anti-guinea pig IgG for 2 hours a room temperature. Cells were washed again with 0.05% Triton-X in PBS prior to the addition of TrueBlue KPL peroxidase substrate, which allows the visualization of infected foci as blue spots. The foci were visualized and counted using an ImmunoSpot CTL Elispot plate reader.

## Peptide library and optimal 8-mer 9-mer epitopes

SARS-CoV-2 and SARS structural protein peptide libraries were obtained from BEI Resources. A SARS spike (catalog NR-2669) peptide library was used in place of SARS-CoV-2 spike due to limited reagent availability. The library consisted of a 169-peptide array of 15-mers to 20-mers overlapping by approximately 10 amino acids and spanning the length of the SARS Urbani strain S protein (GenBank: AY278741). The SARS CoV-2 envelope peptide library (catalog NR-52405) consisted of an array of 10 peptides ranging from 12-mer to 15-mers and overlapping by 10 amino acids spanning the length of the envelope protein of SARS-CoV-2 USA-WA1/2020 strain (GenPept: QHO60596). The SARS-CoV-2 membrane peptide library (catalog NR-52403) consisted of an array of 31 12-mer to 17-mer peptides overlapping by 10

amino acids spanning the length of the membrane protein of SARS-CoV-2 USA-WA1/2020 strain (GenPept: QHO60597). The SARS-CoV-2 nucleocapsid peptide library (catalog NR-52404) consisted of an array of 59 peptides ranging from 13-mers to 17-mers with 10 amino acids of overlap spanning the length of the nucleocapsid protein of SARS-CoV-2 USA-WA1/2020 strain (GenPept: QHO60601). Amino acid sequence information can be found in **S1 Table**. Each peptide came in a lyophilized vial and was reconstituted in 90% DMSO to 10 mg/ml and oriented in a 96 well plate format. During reconstitution, no peptides were noted as insoluble. After reconstitution, subsets of peptides were consolidated to form 11 peptide pools containing various regions or predicted subdomains of each protein (N-terminal region of S1, receptor binding domain, C-terminal region of S1, S2, N-terminal region of nucleocapsid, RNA binding domain of nucleocapsid, nucleocapsid group 3, dimerization domain of nucleocapsid, C-terminal region of nucleocapsid, envelope, and membrane) (**S5 Fig**). In addition, smaller peptide pools were formed by combining analogously oriented wells from each peptide plate (e.g. all A1 peptides are pooled). Once peptide oligomer hits were identified, several potential optimal 8-mer and 9-mer peptides were predicted for each hit based on known $K^b$ and $D^b$ peptide anchor residues [48] and purchased from 21$^{st}$ Century Biochemicals at >90% purity.

## Prediction of $K^b$ and $D^b$ relative binding affinities

To determine the predicted binding affinities of the putative SARS-CoV-2 8-mer or 9-mer CD8+ T cell peptide epitopes to MHC class I, using the NetMHCpan 3.0 server, which uses artificial neural networks to predict relative binding affinities of peptides to any MHC molecule [58]. We utilized the amino acid sequences for SARS-CoV-2 membrane protein or spike protein and prompted the program to predict binding affinities for either 8-mer or 9-mer peptides of these proteins bound to either $K^b$ or $D^b$ molecules.

## RMA-S stabilization assay

To determine the ability of the SARS-CoV-2 8-mer or 9-mer CD8+ T cell peptide epitopes to stabilize either $K^b$ or $D^b$ MHC molecules, we utilized the lymphoma mutant cell line RMA-S cells which are deficient in TAP and lack the ability to process endogenous peptide antigens in an RMA-S stabilization assay [59,60]. RMA-S cells were cultured in complete RPMI medium at 37 degrees Celsius, 5% $CO_2$ until the night before the assay, when the cells were shifted to 29 degrees Celsius, 5% $CO_2$. The cells were then incubated for 4 hours with decreasing concentrations of each peptide at 29 degrees Celsius, and then shifted back to 37 degrees Celsius, 5% $CO_2$ for 1 hour. The cells were then washed with cold PBS and stained for $K^b$ (clone AF6-88.5.5.3) and $D^b$ (clone (28-14-8) molecules at 4 degrees Celsius. The cells were then washed with ice cold PBS and run on an Attune focusing flow cytometer. As positive controls for $K^b$ and $D^b$ stabilizers respectively, Ova peptide (SIINFEKL) and ZIKV peptide $E_{294}$ (IGVSNRDFV) were used [47]. Fluorescence index was calculated by dividing the geometric mean fluorescence intensity (gMFI) of the peptide pulsed cells by non-peptide pulsed cells. Data is displayed as a percentage of the maximum fluorescence index of each peptide serial dilution.

## Statistical analysis

Statistical analyses were performed using Graph Pad Prism. Statistical significance involving serology (AUC analysis and $NT_{50}$ analysis), $T_{fh}$ analysis, T effector analysis, and viral titer analysis was determined by Mann-Whitney test. Statistical significance involving mRNA expression over time or infection over time was determined by 2-way ANOVA. Correlative analysis was performed using linear regression analysis and a two-tailed Pearson analysis.

## Supporting information

**S1 Fig. Protein expression stability from expression constructs.** $2 \times 10^6$ Murine 3T3 cells were transfected with 2 μg of either hACE2 or GFP mRNA and plated at a density of $5 \times 10^6$ cells in each well of a 6 well dish. At 24, 48, and 72 hours GFP expression was evaluated by flow cytometry compared to untransfected cells.
(TIF)

**S2 Fig. Acute phase antibody and boosted $T_{fh}$ response in hACE2 or GFP RNA transfected and SARS-CoV-2 infected mice.** (**A**) Spike receptor binding domain ELISA. Recombinant SARS-CoV-2 spike RBD protein was used to coat an immunosorbent plate. Serum from transfected and infected mice was serially diluted and used to determine RBD binding potential by absorbance at 450nm with increasing serial dilution and area under the curve calculation. (**B**) Neutralization potential of polyclonal sera. Serum from transfected and SARS-CoV-2 infected mice was serially diluted and incubated with ~100 focus forming units of SARS-CoV-2 to allow complexes to form. Virus-serum complexes were then overlaid on a Vero-WHO monolayer and allowed to infect for 24 hours, at which point the plates were fixed and developed (see materials and methods). Neutralization was determined by enumerating a reduction in infectious particles with increased serum concentration and determining the $EC_{50}$. (**C**) $T_{fh}$ gating strategy and frequency. Splenocytes from SARS-CoV-2 infected mice were harvested 5 days post boost and were incubated in Fc block in PBS for 1 hour at 4 degrees Celsius. The cells were then washed with PBS and stained for CXCR5, CD62L, CD8, CD4, PD-1, CD3, B220, and CD4 before being washed with PBS and run on an Attune focusing flow cytometer. $T_{fh}$ cells were defined as lymphocytes based on forward and side scatter, singlets, B220 negative, CD3 positive, CD4 positive, PD-1 and CXCR5 high. Statistical significance was determined by Mann-Whitney test.
(TIF)

**S3 Fig. T cell epitope mapping gating strategy.** T cells were defined by a lymphocyte gate based on size and granularity and CD19 negative. T cells were further classified as CD8+ or CD4+ T cells by staining CD8+/CD4- or CD4+/CD8- respectively. Antigen responsive T cells were defined by IFN-γ expression.
(TIF)

**S4 Fig. Viral burden in Ifnar$^{1-/-}$ mice at day 3 post infection Ifnar$^{1-/-}$ mice were transfected with 10 μg of either GFP or hACE2 RNA.** 24 hours following transfections, mice were infected with $5 \times 10^4$ focus forming units (FFU) of SARS-CoV-2 via IV and IN combination route (100 μl and 20 μl, respectively). n = 6 GFP and n = 7 hACE2 were used to quantify viral burden at 3 days post infection in the lungs (**A**), spleen (**B**), liver (**C**), kidney (**D**), brain (**E**), and whole blood (**F**) by qRT-PCR.
(TIF)

**S5 Fig. Plate maps of SARS-CoV and SARS-CoV-2 peptide libraries.** Peptide libraries spanning the SARS-CoV or SARS-CoV-2 structural proteins were obtained from BEI (**S1 Table**). Every 12-18-mer peptide came in a lyophilized vial and was reconstituted in 90% DMSO to 10 mg/ml and oriented in a 96 well plate format. Subsets of peptides were consolidated to form 11 peptide pools containing various regions or predicted subdomains of each protein (N-terminal region of S1, receptor binding domain, C-terminal region of S1, S2, N-terminal region of nucleocapsid, RNA binding domain of nucleocapsid, nucleocapsid group 3, dimerization domain of nucleocapsid, C- terminal region of nucleocapsid, envelope, and membrane). To aid in identification, peptide pools of 1–5 peptides were also made consisting of the identical

well of each plate (e.g. all A1 wells were pooled).
(TIF)

**S6 Fig. Pooled peptide screen for CD4+ T cell epitope identification. A**) CD4+ T cell responses to pooled peptide domains. Each peptide library was demarcated into peptides contained in functional domains of each protein and peptides contained in each domain were pooled into equimolar pools (11 total pools). 5 days post boosted infection with SARS-CoV-2 following transfection with either hACE2 or GFP mRNA, splenocytes were harvested and stimulated for 6 hours with each domain peptide pool in the presence of brefeldin A. After stimulation, cells were stained for flow cytometry to evaluate the frequency of responsive CD4+ T cells by IFN-γ expression. (**B**) CD4+ T cell responses to smaller "well" peptide pools. Each library was incorporated into multiple 96-well plate formats (**S5 Fig**). Within the same layout, wells from the plates were pooled such that all A1 peptides were pooled, all A2 peptides, etc. maintaining the 96-well plate format, but reducing the overall number of samples that needed to be screened. 5 days post boost following transfection with hACE2 mRNA, splenocytes were harvested and stimulated with each peptide pool in the presence of brefeldin A. The frequency of IFN-γ+ CD4+ T cells is enumerated in a heat map format as the average responses of 3 mice.
(TIF)

**S7 Fig. D$^b$ RMA-S stabilization assay.** To determine relative ability of individual peptide variants to stabilize the D$^b$ molecule, decreasing concentrations of each peptide variant were incubated for 4 hours with TAP deficient RMA-S cells at 29 degrees C before being moved to 37 degrees C for 1 hour. Cells were then stained with anti- D$^b$ APC and geometric mean fluorescent intensity (gMFI) was measured on an Atttune focusing flow cytometer. Fluorescence index (FI) was determined by dividing the gMFI of cells pulsed with peptide by cells with no peptide. Data is presented as a percentage of the maximum FI for each peptide. As a positive control, the D$^b$ restricted peptide ZIKV E$_{294}$ was used.
(TIF)

**S8 Fig. Amino acid alignment of spike of SARS-CoV and spike of SARS-CoV-2.** The amino acid sequences of SARS-CoV-2 Spike (QIG55857.1) and SARS-CoV Spike (ACZ72195.1) were aligned using NCBI based COBALT. Specific regions of interest included, the receptor binding domain (RBD) for SARS-CoV-2 is boxed blue, the location of SARS CoV-2 S1 of spike is noted by orange arrows and S2 is noted by green arrows. Underlined are the SARS-CoV CD4+ and CD8+ multi-mers that were initially screened in the epitope identification studies (**Figs 3 and 4**) and the numbers under each putative epitope are the panels that were screened for the final SARS-CoV-2 CD8+ T cell epitope identification. Bolded are the identified SARS-CoV-2 CD4+ T cell epitopes and in red italics are the identified SARS-CoV-2 CD8+ T cell epitopes.
(TIF)

**S9 Fig. Confirmation of optimal epitope responses in K18-hACE2 transgenic model of infection.** K18-hACE2 transgenic mice were infected with $10^4$ FFU of SARS-CoV-2 (IN route). At day 10 post infection, a mouse was humanely euthanized and splenocytes harvested for peptide stimulation. Splenocytes were stimulated for 6 hours with each peptide in the presence of brefeldin A. After stimulation, cells were stained for flow cytometry to evaluate the frequency of responsive CD8+T cells by IFN-γ and TNF-α expression.
(TIF)

**S1 Table.**
(PDF)

## Author Contributions

**Conceptualization:** Mariah Hassert, James D. Brien, Amelia K. Pinto.

**Data curation:** Mariah Hassert.

**Formal analysis:** Mariah Hassert, Elizabeth Geerling, E. Taylor Stone, Tara L. Steffen.

**Funding acquisition:** Mariah Hassert, James D. Brien, Amelia K. Pinto.

**Investigation:** Mariah Hassert, Elizabeth Geerling, E. Taylor Stone, Tara L. Steffen.

**Methodology:** Mariah Hassert, James D. Brien, Amelia K. Pinto.

**Project administration:** James D. Brien, Amelia K. Pinto.

**Resources:** Mariah Hassert, Madi S. Feldman, Jacob Class, Justin M. Richner, James D. Brien, Amelia K. Pinto.

**Software:** James D. Brien, Amelia K. Pinto.

**Supervision:** James D. Brien, Amelia K. Pinto.

**Visualization:** Mariah Hassert.

**Writing – original draft:** Mariah Hassert, James D. Brien, Amelia K. Pinto.

**Writing – review & editing:** Mariah Hassert, Elizabeth Geerling, E. Taylor Stone, Madi S. Feldman, Alexandria L. Dickson, James D. Brien, Amelia K. Pinto.

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
