## [Decision Letter · Decision Letter 0]

3 Nov 2020

Dear Dr. Pinto,

Thank you very much for submitting your manuscript "mRNA induced expression of human angiotensin-converting enzyme 2 in mice for the study of the adaptive immune response to severe acute respiratory syndrome coronavirus 2" for consideration at PLOS Pathogens. As with all papers reviewed by the journal, your manuscript was reviewed by members of the editorial board and by several independent reviewers. In light of the reviews (below this email), we would like to invite the resubmission of a significantly-revised version that takes into account the reviewers' comments.

We cannot make any decision about publication until we have seen the revised manuscript and your response to the reviewers' comments. Your revised manuscript is also likely to be sent to reviewers for further evaluation.

Sincerely,

Shin-Ru Shih

Section Editor

PLOS Pathogens

Shin-Ru Shih

Section Editor

PLOS Pathogens

Kasturi Haldar

Editor-in-Chief

PLOS Pathogens

orcid.org/0000-0001-5065-158X

Michael Malim

Editor-in-Chief

PLOS Pathogens

orcid.org/0000-0002-7699-2064

Reviewer's Responses to Questions

**Part I - Summary**

Reviewer #1: The use of the mouse as a model for studying viral pathogenesis and determining vaccine efficacy is crucial in understanding viral disease and establishing correlates of immunity. The mouse system is relatively cheap, small, easy to breed and is supported by a large number of reagents. One of the obstacles of using the mouse model for SARS-CoV-2 work is that mice do not express the ideal ACE2 that is amenable for viral infection. Hassert and colleagues have devised a model whereby they transiently transfected IFNR knockout mice with the human ACE2 using an mRNA. The mRNA was administered via an intranasal and intravenous route followed by prime-boost immunization with wildtype SARS-CoV-2. The system is innovative, but given the current data presented, may have limited use for studying aspects of pathogenesis or immunity, save the T-cell response.

Strengths: The use of an mRNA encoding the human ACE2 as a method to introduce it in mice is conceptually great and allows options for researchers who may not have access to other models for working with SARS-CoV-2. Moreover, during times when resources can be limited or strained, other approaches such as this one can be utilized. They were able to use their system to identify CD4+ and CD8+ T-cell epitopes.

Weaknesses: The data presented in the current manuscript demonstrates there are obstacles in making this model a viable alternative to other methods. While in vitro characterization shows the ability of transiently transfected cells to express hACE2, this was not shown in vivo. The lack of good differentiating antibodies (between murine and human ACE2) is noted, but perhaps qPCR could have been performed in the nasal cavity or from lung homogenates to look for transgene expression. Infected hACE2-transfected IFNR KO mice did not induce significantly higher RBD-specific and neutralizing titers over control (GFP-transfected IFNR KO). This also holds true when looking for viral replication – virally-infected hACE2- and GFP-transfected IFNR KO mice did not demonstrate any significant differences. This suggests that the in vivo transfection may not have worked as successfully as anticipated – this should be addressed. The authors should try to figure the efficiency by which their mRNA administration in vivo is working. Lastly, while the authors were able to identify T-cell epitopes, that responding hACE2 mice had a large standard deviation, which suggest that transfection efficiency of the mRNA-ACE2 – however this is speculative on the reviewers part.

Line 123: It might be beneficial for the rest of the community if the authors list the antibodies that were used to attempt to differentiate human and murine ACE2 in the methods section. This can save others effort and funds.

Line 151: It would be useful to know the concentration or the amount of mRNA that was used for each route of administration (intravenous and intranasal). Is there a rationale why both administrations were utilized instead of one or the other. Our own experience with using the Polyplus in vivo-jet RNA transfection system suggests that transgene expression via the intranasal route was inferior (metric was protection against a lethal viral challenge) to a systemic intramuscular route of administration. Have the authors administered the mRNA either intranasal only or intramuscular only and compare the efficiency of gene expression? It may benefit the readers to know whether one route of administration is sufficient instead of using both routes.

Line 369: The authors should also include in the discussion the possibility that the intranasal administration of mRNA was not efficient in delivering the mRNA encoding hACE2. Have the authors assessed the optimal amount of RNA delivered? This and the points discussed by the authors may contribute to not detecting any measurable viral replication or pathogenesis in the current experimental model. Lastly, the results from this study demonstrates that there is a robust B- and T-cell response to SARS-CoV-2 infection –but it could be reflective of the route of RNA administration that is most efficient in delivery (IV vs IN). This is speculative; however, this may help explain the current results.

Reviewer #2: Using an mRNA delivery platform to induce expression of hACE2 in mice, Hassert et al. investigate the adaptive immune response to SARS-CoV-2 in type I interferon deficient mice. They demonstrate that high affinity antibody response to SARS-CoV-2 is independent of the hACE2 expression, but they found a more robust T cell response when hACE2 mRNA was provided to mice. The authors further profile and define CD8+ and CD4+ structural peptide epitopes that are targeted during infection, and find similar targets as to what is seen in humans.

Unfortunately, beyond an indirect T cell response, there is no evidence that this mouse model system induced infection. The paper as framed suggests otherwise. Further, there is no evidence of mRNA expression of hACE2 in the actual mouse model, the paper also suggests otherwise. Validation of mRNA expression was done in cell culture not in vivo.

**Part II – Major Issues: Key Experiments Required for Acceptance**

Reviewer #1: The authors should show the efficiency by which the mRNA is being delivered intranasally or systemically. This will allow them to figure out the viability of their model being susceptible to SARS-CoV-2 infection or immunization.

Reviewer #2: Unfortunately, beyond an indirect T cell response, there is no evidence that this mouse model system induced infection. The paper as framed suggests otherwise. Further, there is no evidence of delievered mRNA expression of hACE2 in the actual mouse model, the paper also suggests otherwise. Validation of mRNA expression was done in cell culture not in vivo.

**Part III – Minor Issues: Editorial and Data Presentation Modifications**

Reviewer #1: (No Response)

Reviewer #2: Figure 1C and 1D have no error bars? Since infection initiation only probed at 24H, expression levels should be assessed at 24H.

No significant differences in viral loads in mice or any infectious virus found, yet genome copies were found in all organs. Authors should use non-infected control mouse group to confirm primer specificity.

Robust T cell response compared to what? Need positive control here (ie transgenic hACE2 mouse).

PLOS authors have the option to publish the peer review history of their article (what does this mean?). If published, this will include your full peer review and any attached files.

Reviewer #1: No

Reviewer #2: No
---

## [Editor Report · Decision Letter 1]

2 Dec 2020

Dear Dr. Pinto,

We are pleased to inform you that your manuscript 'mRNA induced expression of human angiotensin-converting enzyme 2 in mice for the study of the adaptive immune response to severe acute respiratory syndrome coronavirus 2' has been provisionally accepted for publication in PLOS Pathogens.

Best regards,

Shin-Ru Shih

Section Editor

PLOS Pathogens

Shin-Ru Shih

Section Editor

PLOS Pathogens

Kasturi Haldar

Editor-in-Chief

PLOS Pathogens

orcid.org/0000-0001-5065-158X

Michael Malim

Editor-in-Chief

PLOS Pathogens

orcid.org/0000-0002-7699-2064
---

## [Editor Report · Acceptance letter]

9 Dec 2020

Dear Dr. Pinto,

We are delighted to inform you that your manuscript, "mRNA induced expression of human angiotensin-converting enzyme 2 in mice for the study of the adaptive immune response to severe acute respiratory syndrome coronavirus 2," has been formally accepted for publication in PLOS Pathogens.

Best regards,

Kasturi Haldar

Editor-in-Chief

PLOS Pathogens

orcid.org/0000-0001-5065-158X

Michael Malim

Editor-in-Chief

PLOS Pathogens

orcid.org/0000-0002-7699-2064